# Deciphering Promoter Hypermethylation of Genes Encoding for RASSF/Hippo Pathway Reveals the Poor Prognostic Factor of RASSF2 Gene Silencing in Colon Cancers

**DOI:** 10.3390/cancers13235957

**Published:** 2021-11-26

**Authors:** Marc Riffet, Yassine Eid, Maxime Faisant, Audrey Fohlen, Benjamin Menahem, Arnaud Alves, Fatéméh Dubois, Guénaelle Levallet, Céline Bazille

**Affiliations:** 1Department of Pathology, CHU de Caen, 14000 Caen, France; riffet-m@chu-caen.fr (M.R.); faisant-m@chu-caen.fr (M.F.); fatemeh.dubois@unicaen.fr (F.D.); 2Normandie Université, UNICAEN, CEA, CNRS, ISTCT, CERVOxy Group, GIP CYCERON, 14074 Caen, France; fohlen-a@chu-caen.fr; 3Polyclinique du Parc, 14000 Caen, France; eidyassine@hotmail.fr; 4Normandie Université, UNICAEN, INSERM UMR 1086, ANTICIPE, 14000 Caen, France; menahem-b@chu-caen.fr (B.M.); alves-a@chu-caen.fr (A.A.); 5Department of Radiology, CHU de Caen, 14000 Caen, France; 6Department of Digestive Surgery, CHU de Caen, 14000 Caen, France; 7Structure Fédérative D’oncogénétique cyto-MOléculaire du CHU de Caen (SF-MOCAE), CHU de Caen, 14000 Caen, France

**Keywords:** colon cancers, RASSF/Hippo pathway, molecular biomarker

## Abstract

**Simple Summary:**

Colorectal cancer (CRC) is a major public health issue due to its incidence and mortality. Thus, the development of molecular biomarkers is essential to optimize its therapeutic management. Such markers could be identified among the members of the RASSF/Hippo pathway. Indeed, epigenetic alterations are strongly implicated in colorectal carcinogenesis and this pathway is altered in many cancers, mainly by hypermethylation of the promoter of the gene coding for its members. The objectives of the study were to map the hypermethylation of the RASSF/Hippo pathway promoters in a morphologically, clinically, and prognostically well-characterized population of colon cancers. This first report of a whole systematic analysis of the Hippo pathway in colon cancer highlights *RASSF2* gene promoter hypermethylation as a worst prognostic factor and a tool to be sought in clinical practice to improve therapeutic management.

**Abstract:**

The aims of this study were to assess the frequency of promoter hypermethylation of the genes encoding the Ras associated domain family (RASSF)/Hippo pathway, as well as the impact on overall (OS) and progression-free survival (PFS) in a single-center retrospective cohort of 229 patients operated on for colon cancers. Hypermethylation status was investigated by methylation-specific PCR on the promoters of the *RASSF1/2*, *STK4/3* (encoding Mammalian Ste20-like protein 1 and 2 (MST1 and 2), respectively), and *LATS1/2* genes. Clinicopathological characteristics, recurrence-free survival, and overall survival were analysed. We found the RASSF/Hippo pathway to be highly silenced in colon cancer, and particularly RASSF2 (86%). The other promoters were hypermethylated with a lesser frequency of 16, 3, 1, 10 and 6%, respectively for *RASSF1*, *STK4*, *STK3*, *LATS1*, and *LATS2* genes. As the hypermethylation of one RASSF/Hippo family member was by no means exclusive from the others, 27% of colon cancers displayed the hypermethylation of at least two RASSF/Hippo member promotors. The median overall survival of the cohort was 60.2 months, and the median recurrence-free survival was 46.9 months. Survival analyses showed a significantly poorer overall survival of patients when the *RASSF2* promoter was hypermethylated (*p* = 0.03). The median OS was 53.5 months for patients with colon cancer with a hypermethylated *RASSF2* promoter versus still not reached after 80 months follow-up for other patients, upon univariate analysis (HR = 1.86, [95% CI: 1.05–3.3], *p* < 0.03). Such difference was not significant for relapse-free survival as in multivariate analysis. A logistic regression model showed that *RASSF2* hypermethylation was an independent factor. In conclusion, *RASSF2* hypermethylation is a frequent event and an independent poor prognostic factor in colon cancer. This biomarker could be investigated in clinical practice.

## 1. Introduction

Colorectal carcinogenesis results from an accumulation of genomic abnormalities and epigenetic deregulations. These genomic and epigenetic alterations occur through three main mechanisms: chromosomal instability, microsatellite instability, and CpG Island Methylator Phenotype (CIMP) hypermethylation [1,2]. This CIMP phenotype is characterised by cytosine hypermethylation in the promoter regions of tumour suppressor genes, resulting in the inactivation of the expression of these genes [2,3,4]. Colorectal cancer (CRC) is the third most common cancer worldwide and the second most common cancer death regardless of age or gender. The overall 5- and 10-year survival is estimated at 62% and 50% in men and 64% and 54% in women, respectively, for all stages [5,6].

The RASSF/Hippo signalling pathway is composed of several members: (i) regulators, such as members of the Ras associated domain family (RASSF), (ii) kinases, constituting the core of this pathway: Mammalian Ste20-like protein 1 and 2 (MST1 and 2), Large Tumor Suppressor 1 and 2 (LATS1 and 2), and Nuclear Dfb2- Related 1 and 2 (NDR1 and 2) kinases, and (iii) transcriptional cofactors: Yes-associated protein (YAP) and transcriptional coactivator with PDZ-binding motif (TAZ). In a healthy cell, Hippo pathway regulators interact with MST1/2 kinases and regulate their activity [7]. When activated by phosphorylation, MST1/2 then phosphorylate the kinases LATS1/2 and NDR1/2. In response, these kinases phosphorylate YAP and TAZ [7,8]. Phosphorylation of these transcriptional cofactors leads to their intra-cytoplasmic sequestration, ubiquitination, and destruction by the proteasome [9]. When one of the members of the RASSF/Hippo pathway is inactivated, usually by the hypermethylation of its promoter [9], YAP and TAZ are then abnormally activated (under their dephosphorylated form) and translocate into the nucleus to activate transcription of genes related to cell proliferation, migration, and resistance to apoptosis [9,10,11,12,13].

The *RASSF2* gene, located at 20p13, encodes three isoforms named RASSF2A through C. The A and C isoforms exhibit a Ras association domain, like RASSF1A [14]. RASSF2A is primarily nuclear-localised; its interaction with MST1/2 kinases allows its cytoplasmic translocation and activation by phosphorylation of MST1/2 kinases [14,15]. The *RASSF2* gene promoter is frequently hypermethylated in many types of cancers, such as bronchopulmonary, gastric, colorectal, breast, endometrial, and upper aerodigestive cancers [15]. The hypermethylation status of the *RASSF2* promoter could be a poor prognostic factor, as its hypermethylation would promote tumour cell aggressiveness [14].

YAP and TAZ proteins, in dephosphorylated form, preferentially interact with TEA Domain family members (TEAD) [13]. The canonical regulatory pathway is represented by NDR kinases. In contrast, the non-canonical regulatory pathway involves other mechanisms such as cell density, extracellular matrix stiffness, hypoxia, or activation of G protein-coupled receptors: RhoA and RhoB [13,16,17,18]. Abnormal activation of YAP/TAZ has been implicated in the tumourigenesis of several cancers, and this deregulation induces enhanced migratory and invasive abilities of tumour cells [13,19].

Literature data concerning alterations in the RASSF/Hippo pathway in CRC are still patchy and sometimes controversial. Some arguments suggest that alterations in this pathway may be involved in the transformation and spreading of colonic tumour cells, as observed in other cancers. Indeed, it has been described that the overexpression of YAP in colon cancer promotes tumour cell proliferation, distant metastasis, and is an independent prognostic factor for patient survival [20]. This overexpression could result from the inactivation of one of the members of the RASSF/Hippo pathway by promoter hypermethylation. Promoter hypermethylation of the *RASSF1A* and *RASSF2* genes has been described in colorectal cancers in 15–45% and 42–70% of cases, respectively [14,21,22]. The *LATS1* gene promoter is hypermethylated in 57% of cases [23]. Furthermore, the kinases can be inactivated by microRNAs; miR-590-3p for LATS1 or miR-103 for *LATS2* [24,25].

The objectives of this study were to evaluate the frequency of the promoter methylation of genes involved in the RASSF/Hippo pathway in colon cancer, as well as its impact on overall and recurrence-free survival in a cohort of colon cancer patients.

## 2. Materials and Methods

### 2.1. Patients

We selected a retrospective population of patients operated on for colon cancer at the Caen University Hospital between January 2010 and September 2013. Two hundred and twenty-nine patients were selected continuously regardless of the stage of the disease. All patients with rectal cancer were excluded due to the use of neo-adjuvant radiotherapy, which can cause genomic and molecular alterations in tumour cells. Clinical data were collected for each patient. This information concerns the location of the tumour, the date of surgery, the character of occlusion or not, the presence of synchronous hepatic or extra-hepatic metastasis, the occurrence of a recurrence, with its date of discovery and possible treatment. The administration of adjuvant chemotherapy, first-, second-, or third-line adjuvant chemotherapy, and the different therapies used were also collected. Finally, in order to establish overall and recurrence-free survival data, the date of last news and the date of death were collected, with a time point set at 30 June 2018.

Morphological criteria of interest were collected on the macroscopic report, and all histological criteria were analysed and reviewed by two pathologists (CB and MR) on the histological slides stained with Safran Hematoxylin Eosin. The different histological criteria are listed in table X, along with concern size, grade of differentiation, pTNM stage according to the 2017 UICC classification (8th edition), presence or absence of lymphatic and/or venous tumour emboli, presence or absence of tumour deposits, budding grade, the intensity of inflammatory infiltrate, and pushing or infiltrating type invasion front. MisMach Repair (MMR) status was analysed by the immunohistochemical technique with the following four antibodies: M1 (anti-MLH1), G219-1129 (anti-MSH2), 44 (anti-MSH6), and EPR3947 (anti-PMS2) on a Ventana Benchmark ULTRA automated system. In the case of conserved nuclear expression throughout the tumour proliferation, the lesion was classified as proficient MMR in immunohistochemistry (pMMR-IHC). In the case of the extinction of one or two markers of the MMR system in immunohistochemistry, an additional microsatellite instability test was performed using molecular biology to confirm the deficient MMR (dMMR) status. The search for somatic mutations on codons 12 and 13 of exon 2 of the KRAS gene, and on codon 600 of the BRAF gene, was carried out at the Laboratoire de Biologie et Génétique du Cancer of the Centre François Baclesse by pyrosequencing.

As required by French laws, all patients provided informed consent, and the study was approved by the institutional ethics committee (North-West-Committee-for-Persons-Protection-III N°DC-2008-588).

### 2.2. DNA Extraction and Methylation-Specific PCR Assay

DNA extraction was performed with the Maxwell^®^ RSC DNA FFPE kit. The extracted DNAs were treated with sodium bisulfite using the EpiTec^®^Bisulfite kit (QIAGEN, Hilden, Germany). The bisulfonation reaction was performed in a thermocycler with 3 cycles of denaturation and incubation. The converted DNAs were purified in the presence of a “BL” buffer, transferred to a column, and incubated in the presence of a “BD” desulfonation buffer. For each DNA, two amplification reactions were performed to determine the presence or absence of the methylation of the promoter under consideration: either with a pair of primers recognising unmethylated sequences (U) or with a pair of primers recognizing methylated sequences (M) [18,26,27], (Appendix A). The amplicons were separated on a 3.5% agarose gel enriched with GelRed. The positive methylation control used was a commercial “methylated” DNA (cpGenome Universal methylation DNA, S7821 QBiogen, Tamil Nadu, India). The no-methylation control was lymphocyte DNA. A negative control was systematically performed for each PCR.

### 2.3. Statistical Analysis

Qualitative and quantitative variables were compared using the Chi2 or Student’s *t*-test, respectively (Mann–Whitney test and Fisher’s exact test when the validity conditions of Student’s and Chi2 tests were not verified). Survival was analysed using Kaplan–Meier curves. A univariate and multivariate Cox model was performed to identify prognostic factors for overall survival and recurrence-free survival. Variables that were associated with the predictable variable (survival) were selected using a backward selection process to produce the final multivariate model. A logistic regression model with a backward selection process was used to search for an association between the RASSF2 marker and the prognostic criteria (outcomes) was used in the survival model. The statistical difference was considered significant when *p* was less than 0.05. All statistical analyses were performed using SAS 9.4 software (SAS Institute, Cary, NC, USA) by Y.E.

## 3. Results

### 3.1. Clinical and Histo-Prognostic Population Characteristics

The clinical criteria are detailed in Table 1. The average age of the cohort was 71 years, with extremes ranging from 27 to 99 years. One hundred and twenty patients were male, representing 52.4% of the study cohort. The tumour was located on the right in 52.0% of cases and on the left in 48.0% of cases. Stage I patients constituted 9.1% of the patients in the cohort, 35.4% for stage II patients, 38.0% for stage III patients, and 17.5% for stage IV patients.

Among the 229 patients included in this study, no patient was lost to follow-up. During the study, 96 patients died. One hundred and eleven patients were alive after 5 years, giving an overall 5-year survival of 48.5%. The overall and recurrence-free survival of patients showed a median overall survival of 60.2 months and a median recurrence-free survival of 46.9 months. Histological and molecular criteria are detailed in Table 2 and Appendix A. Of the 229 colic adenocarcinomas analysed, most were low grade: well-differentiated (44.1%). Twenty-nine tumours were mucinous adenocarcinoma (12.7%). In the majority of cases (61.1%), the tumours were stage pT3. The pT1 and pT2 stages represented 3.0 and 7.8%, respectively.

Fifty-three tumours (22.2%) ulcerated the colic serosa (stage pT4a), and two (0.9%) invaded adjacent organs (stage pT4b). One hundred and thirteen patients (49.3%) had no lymph node metastases. We observed the presence of tumour emboli in 96 patients (41.9%). Microsatellite phenotype analysis by immunohistochemistry was performed in 14.8% of cases with a dMMR phenotype for 10 tumours. Thirty-four patients (42.5% of cases analysed) had a KRAS or NRAS mutation, and six had a BRAF mutation (8% of cases analysed).

### 3.2. MS-PCR Results and Impact on Survival

We first investigated the methylation of all the promoters of the genes involved in the RASSF/Hippo pathway, specifically RASSF1/2, STRK4/3 (coding for MST1 and MST2 proteins respectively), and LATS1/2, by MS-PCR using DNA extracted from the tumour blocks of 100 patients. We found the RASSF/Hippo pathway to be highly silenced in colon cancer, and particularly RASSF2 (86%). The other promoters were hypermethylated with a lesser frequency of 16, 3, 1, 10 and 6%, respectively, for *RASSF1*, *STK4*, *STK3*, *LATS1,* and *LATS2* genes. As the hypermethylation of one RASSF/Hippo family member was by no means exclusive from the others, 27% of colon cancers displayed hypermethylation of at least two RASSF/Hippo member promotors (Figure 1).

In addition, we found a tendency for poorer prognosis when the *RASSF2* promoter was hypermethylated, whereas the hypermethylation of *LATS2* promoter showed a trend for a better prognosis, but not significantly (data not shown). We, therefore, subsequently included 129 additional patients and studied only the methylation status of the *RASSF2* and *LATS2* promoters. The inclusion of these supplementary patients allowed us to demonstrate the significantly poorer overall survival of patients when the *RASSF2* promoter was methylated (*p* = 0.03). The median OS was 53.5 months for patients with colon cancer with a hypermethylated RASSF2 promoter versus still not reached after 80 months follow-up for other patients, upon univariate analysis (HR = 1.86, [95% CI: 1.05–3.3], *p* < 0.03), (Figure 2). However, there was no significant difference in recurrence-free survival (*p* = 0.30). Furthermore, the extension of the cohort failed to find a significant association between *LATS2* methylation and the overall, or relapse-free survival, of patients.

We analysed the overall and recurrence-free survival using a Cox model, in univariate and multivariate analysis, on the cohort of 229 patients. However, we could not find any significant impact of the hypermethylation of the promoter of the *RASSF2* gene. 

Finally, a logistic regression model to search for an association between the hypermethylation of the promoter of the *RASSF2* gene and the prognostic criteria used in the survival model showed that *RASSF2* gene promoter hypermethylation was an independent factor (Table 3).

Likewise, by classifying tumour samples from patients in this study on their inflammatory component (no inflammation, moderate inflammation, significant inflammation, presence of a Crohn-like inflammatory response), we report that this variable was statistically independent of *RASSF2* gene promoter hypermethylation status.

Finally, we questioned the TCGA database to test if *RASSF2* gene promoter hypermethylation predicts worse overall survival in the 229 patients with colon cancer from our collection is consistent in another cohort. Analysis of the influence of the expression level of RASSF2 mRNA was performed in 438 patients with colon cancer. Patients were classified according to whether their tumour strongly (*n* = 336) or weakly (*n* = 102) expresses the RASSF2 mRNA. Survival analysis reveals that the low RASSF2 mRNA expression (possibly due to hypermethylation of the RASSF2 promoter) predicts significant worse overall survival in 438 patients with colon cancer (*p* score: 0.038, expression of RASSF2 in colorectal cancer-The Human Protein Atlas, Appendix A).

## 4. Discussion

Colon cancers are a public health issue due to their incidence and mortality. It is therefore essential to determine histological and molecular markers to optimise the therapeutic management of patients. The objective of our study was to analyse the impact of the alteration of the members of the RASSF/Hippo pathway in patients with colon cancer, with all stages included. Our results show that the different members of this pathway, RASSF1/2, MST1/2, and LATS1/2, can be altered by the promoter hypermethylation of their genes with variable frequencies. Nevertheless, it is difficult to compare our frequencies with the data of the literature since alterations in this pathway have never been characterised in a systematic way.

Among the genes studied, we observed that the promoter of the *RASSF2* gene is the most frequently hypermethylated. It is described that the *RASSF2* promoter is frequently hypermethylated in many types of cancers, such as bronchopulmonary, gastric, colorectal, breast, endometrial, and upper aerodigestive cancers [14]. We found an 87.2% frequency of RASSF2 promoter hypermethylation in our cohort, which is consistent with the results observed by Park et al. and Hesson et al., who observed RASSF2 hypermethylation in 72.6 and 70% of cases, respectively [26,28]. However, Harada et al. and Akino et al. observed lower hypermethylation frequencies, 46% and 42%, respectively [19,29]. Furthermore, Park et al. also observed that *RASSF2* promoter hypermethylation was an early-onset event, as of all 16 colorectal adenomas analysed, all had *RASSF2* promoter hypermethylation [28]. Due to the absence of *RASSF2* promoter hypermethylation in normal colonic tissue already reported in the literature [26,28], and of the absence of such hypermethylation in five mucosae that we have tested at the beginning of this study (data not shown), we did not analyse *RASSF2* promoter hypermethylation outside the tumour for all patients in our cohort.

By studying the impact of the methylation status on the 5-year survival of patients in our cohort, we observed that patients with *RASSF2* promoter hypermethylation have a significantly worse prognosis. To date, there is very little data in the literature regarding the impact of *RASSF2* promoter hypermethylation status on survival in CRC patients. The Human Protein Atlas (HPA), which catalogues survival data for many cancers based on the expression level of various proteins, shows that patients with low RASSF2-expressing colon cancer have a poorer prognosis (*p* = 0.038), which is consistent with our results, as promoter methylation of the gene inhibits its expression [29]. However, these data relate to the protein expression of RASSF2, and low protein expression may be related to other causes than the hypermethylation of the *RASSF2* promoter. Other arguments are in favour of a prognostic role of *RASSF2* promoter hypermethylation status. Indeed, *RASSF2* hypermethylation would promote tumour cell aggressiveness, as observed by Luo et al. and Aydin et al. in patients with gastric adenocarcinoma [30,31]. On the other hand, in colon tumour lines, Carter et al. showed that miR-200 targeted *RASSF2* mRNA with the inhibition of its expression, and tumour cells overexpressing this microRNA proliferated more [32]. Furthermore, in CRC, hypermethylation of the *RASSF2* promoter appears to be more frequently associated with the presence of a KRAS mutation [19,28,29,32], although Hesson et al. instead describe an inverse relationship between the frequency of *RASSF2* hypermethylation and the frequency of KRAS mutation [26].

Regarding the methylation status of *RASSF1*, we observed a methylation frequency of 16% in the first 100 patients of our cohort. Hu et al., by conducting a meta-analysis on the hypermethylation status of the *RASSF1* promoter in CRC from 21 studies, reported a highly variable hypermethylation frequency, ranging from 15.84% to 93.3%. However, they note that some studies had *RASSF1* hypermethylation in non-tumour control samples [33]. *RASSF1* promoter hypermethylation is also reported in osteosarcoma, breast cancer, upper aerodigestive tract cancer, glioma, clear cell carcinoma of the kidney, urothelial carcinoma, thyroid carcinoma, and neuroblastoma [34]. Hu et al. also analysed three other studies, which focused on the prognostic impact of *RASSF1* hypermethylation status and highlighted that *RASSF1* hypermethylation is a poor prognostic factor [33]. This prognostic impact has also been observed by Jiang et al., who demonstrated poorer survival of breast cancer patients when the *RASSF1* promoter is hypermethylated [35]. In our study, we could not demonstrate an impact of *RASSF1* hypermethylation on the 5-year survival of patients, possibly due to an insufficient number of patients. Unlike *RASSF2*, *RASSF1* promoter hypermethylation seems to correlate with the absence of KRAS mutation [36]. In addition, Sun et al. observed better chemosensitivity to oxaliplatin when the *RASSF1* promoter is not hypermethylated [37].

We observed a low frequency of promoter methylation of *STK4* and *STK3* (encoding MST1 and MST2, respectively). The alteration of these two genes is not often studied in CRC. Nevertheless, some studies have observed that CRC, with the inactivation of MST1 or MST2, seems to have a worse prognosis. Minoo et al. demonstrated, by studying the expression of MST1 by immunohistochemistry, that the loss of cytoplasmic expression of MST1 was correlated with an advanced TNM stage, the presence of vascular emboli and lower overall survival. However, they did not observe an association between the loss of cytoplasmic MST1 expression and *STK4* promoter hypermethylation [38]. The association between loss of MST1 expression and poorer prognosis has been observed in other types of cancers, including malignant mesothelioma [39]. Furthermore, Guo et al. observed in diffuse gliomas that the *STK4* promoter could be hypermethylated and its hypermethylation promoted tumour cell aggressiveness [40].

We also observed that the *LATS1* and *LATS2* promoters are infrequently hypermethylated, 10% and 6%, respectively. *LATS2* promoter hypermethylation tended to be a better prognostic but not significantly. The HPA atlas also showed that in colonic cancers, LATS2 methylation appeared to have a better prognosis (*p* = 0.018) [41]. However, low expression of LATS2 may be related to other mechanisms than the hypermethylation of its promoter, as observed by Peng et al. Conversely, they observed that low expression of LATS2 is associated with worse prognosis in CRC. Furthermore, miR-372-3p, frequently overexpressed in CRC, targets LATS2 RNA, and its overexpression is correlated with tumour cell aggressiveness [42]. As for LATS1, Wierzbicki et al. observed that its expression is frequently inactivated (89.4%), by hypermethylation of the *LATS1* promoter in more than half of the cases, in contrast to the frequency of methylation that we observed in our study [43]. In other cancer types, LATS1 and LATS2 expression can be decreased by the hypermethylation of their promoter, as reported in breast cancer, soft tissue sarcoma, gastric cancer, non-small cell lung carcinoma, clear cell carcinomas of the kidney, and nasopharyngeal, and oral cavity squamous cell carcinomas, or by loss of heterozygosity, as observed in ovarian, cervical, breast, and liver cancers [11,44,45,46,47]. Inactivation of LATS1 and LATS2 would be associated with more aggressive tumours, and with a poorer prognosis [11], with the exception of oligodendrogliomas, for which hypermethylation of the *LATS2* promoter predicts a better survival, as observed in CRC [48].

## 5. Conclusions

The RASSF/Hippo signalling pathway is altered in many cancers by the hypermethylation of the promoter of genes involved in this process. We studied the impact of promoter hypermethylation of genes involved in the RASSF/Hippo pathway, namely *RASSF1/2*, *STK4/3*, *and LATS1/2*, by studying their methylation status by MS-PCR in a retrospective CRC cohort. Our results show that the hypermethylation of the *RASSF2* gene promoter is frequent and associated with poorer survival in 229 patients with colon cancer. Therefore, there may be a clinical interest in studying the hypermethylation status of the *RASSF2* promoter in CRC tumours as a prognostic factor in order to adapt the therapeutic management.

## Figures and Tables

**Figure 1 cancers-13-05957-f001:**
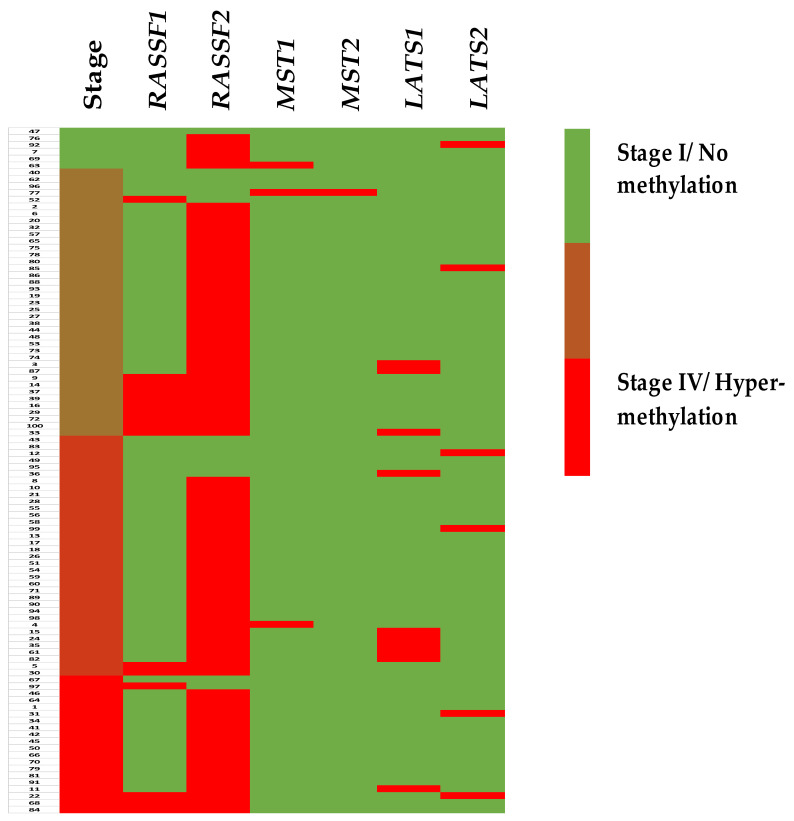
Heat map of RASSF/Hippo signaling pathway data decryption by staging.

**Figure 2 cancers-13-05957-f002:**
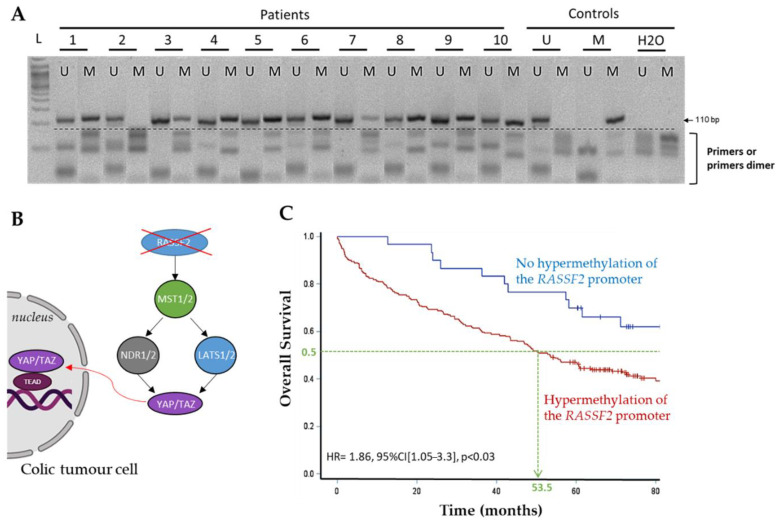
*RASSF2* gene promoter hypermethylation predicts worse overall survival in 229 patients with colon cancer. (**A**) Methylation analysis in 10 colon cancer cases. Lane U: amplified product with primers recognizing unmethylated sequence; Lane M: amplified product with primers recognizing methylated sequence. Control “M”: positive control for hypermethylation; Control “U”: positive control for unmethylation (lymphocyte DNA); H_2_O: negative control. The ladder (L) saw at the first lane is 50 bp. (**B**) Diagram of the consequences of the silencing of *RASSF2* on the nuclear expression of YAP and TAZ in colonic cancer cells. (**C**) *RASSF2* gene promoter hypermethylation impact on overall survival in patients with colon cancer.

**Table 1 cancers-13-05957-t001:** Clinical characteristics of the cohort.

Clinical Characteristics	All Patients N = 229	
	N	%
Age (years old)	71 (27–99)	
Sex (female/male)	109/120	47.6/52.4
Location (right/left)	119/110	52.0/48.0
Tumoral size (cm)	5.05 (0.4–16)	
Clinical stage	21/81/87/40	9.1/35.4/38.0/17.5
I/II/III/IV		
Intestinal occlusion	67	29.2
Synchronous metastasis (no/hepatic/extrahepatic)	188/35/6	82.1/15.3/2.6
Recurrence		
No	180	78,6
Local/Hepatic/Extrahepatic	18/19/8	7.9/8.3/3.5
Death	96	40.6

**Table 2 cancers-13-05957-t002:** Histological characteristics of the cohort.

Histological Characteristics	All Patients N = 229	
	N	%
Differentiation Well/Moderately/Poor Mucinous	101/76/23 29	44.1/33.2/10.0 12.7
Stage T 1/2/3/4a/4b	7/18/140/51/2	3.0/7.8/61.1/22.2/0.9
Stage N 0/1a/1b/1c/2	113/34/37/9/36	49.3/14.9/16.2/3.9/15.7
Emboli	96	41.9
MMR (pMMR/dMMR/NA ^1^)	24/10/195	10.5/4.4/85.1
RAS (no, mutated/NA ^1^)	46/34/149	20.1/14.8/65.1
BRAF (no, mutated/NA ^1^)	69/6/154	30.1/2.6/67.3
*RASSF2* hypermethylation	199	86.9

^1^ Not Available.

**Table 3 cancers-13-05957-t003:** Logistic regression model between histopronostic factors and RASSF2 methylation status.

Clinical and Histological	Univariate Model		
Characteristics	HR	IC 95%	*p*
Age (<70 years old)	2.07	[0.95–4.50]	0.07
Sex (male)	1.98	[0.88–4.44]	0.10
Location (right)	0.57	[0.26–1.25]	0.16
Synchronous metastasis (no metastasis)	1.07	[0.38–2.98]	0.90
Tumoral deposit	2.00	[0.73–5.50]	0.18
Lymphatic tumor emboli	0.94	[0.43–2.03]	0.86
Budding	1.25	[0.48–3.25]	0.64

## Data Availability

All data are stored at the Caen University Hospital center and can be made available upon request.

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
