# Peer review of "Deciphering Promoter Hypermethylation of Genes Encoding for RASSF/Hippo Pathway Reveals the Poor Prognostic Factor of RASSF2 Gene Silencing in Colon Cancers"

_cancers, 2021, doi:10.3390/cancers13235957_

Round 1

Reviewer 1 Report

The authors analyzed promoter hypermethylation of genes encoding for RASSF/Hippo pathway in colon cancers and found that RASSF2 was highly silenced, which was relevant to poorer overall survival of patients. However, more evidences should be provided to support the current conclusion. They should compare promoter hypermethylation of cancer tissues with adjacent normal tissues and between cancers of different histologic grade. In addition, the authors should show the relevance between activity of Yap signaling and promoter hypermethylation of genes encoding for RASSF/Hippo pathway. Results described in the last paragraph should be included.

Author Response

The authors analyzed promoter hypermethylation of genes encoding for RASSF/Hippo pathway in colon cancers and found that RASSF2 was highly silenced, which was relevant to poorer overall survival of patients. However, more evidences should be provided to support the current conclusion.

We would like to thank the reviewer#1 for his/her interests and valuable comments on this manuscript.

They should compare promoter hypermethylation of cancer tissues with adjacent normal tissues and between cancers of different histologic grade.

We understand the concern of the reviewer#1. However, the previous studies by Hesson and Park have already reported the absence of methylation in the adjacent normal tissues of colonic cancer tissues. Indeed:

  • According to Hesson (https://doi.org/10.1038/sj.onc.1208566): “Methylation was observed in 21/30 (70%) tumours. Moreover, this methylation was always tumour-specific and was not detected in the matched patient’s DNA from normal mucosa (taken >10 cm from the primary).”
  • According to Park (https://doi.org/10.1002/ijc.22276.):” We also examined the methylation status in 17 corresponding normal tissues (Fig. 2a). MSP analysis showed that RASSF2A was unmethylated in most corresponding normal tissues, but 2 normal tissues showed weak methylation. These results were further confirmed by cloning and bisulfite sequencing; specifically, only 1–5 CpG sites of a total of 25 CpG sites were methylated in normal tissues (Fig. 2b).”

At the beginning of our study, we performed analyses on 5 normal mucosae far from the tumor. We found that these mucosae were all found to have an unmethylated promotor of the RASSF2 gene (data not shown). Because of the absence of RASSF2 methylation in normal colonic tissue, which has already been reported in the literature and of the absence of such hypermethylation in the 5 mucosa that we have tested, we did not further analyze RASSF2 methylation outside of the tumor (in healthy tissue) of all the patient of our cohort.

We now justify this in the Discussion section as follow:” Because of the absence of RASSF2 promoter hypermethylation in normal colonic tissue already reported in the literature [26, 28] and of the absence of such hypermethylation in five mucosa that we have tested at the beginning of this study (data not shown), we did not analyze RASSF2 promoter hypermethylation outside the tumor for all patient of our cohort.”

In addition, the authors should show the relevance between activity of Yap signaling and promoter hypermethylation of genes encoding for RASSF/Hippo pathway.

We fully agree with this reviewer's remark.

To answer this question, we attempted to perform YAP immunostaining. Indeed, as this reviewer knows perfectly well, YAP is relocated in the nucleus of cells (in particular cancerous cells) when it is active, but this approach has had two limitations: 1) First, the immunostaining of YAP has been found to be very complicated to interpret. The immunostaining of YAP was indeed positive in both the cytoplasm and the nucleus of most cases of our cohort, which hampered the interpretation of the nuclear presence of YAP (by the pathologist or a digital analysis, based on artificial intelligence, an example is available in supplementary data only for reviewer) and therefore the evaluation of variations in the nuclear expression of YAP between patients in this study. As a result of technical difficulties, we failed to show any correlation between “nuclear” YAP expression and promotor hypermethylation 2) Secondly, the immunohistochemistry is anyway limited to answer the question of the activity of YAP, since it is accepted that YAP can sometimes be nuclear although inactive (recently: https://www.cell.com/developmental-cell/fulltext/S1534-5807(21)00726-7).

We would like to answer this question through different technological approaches such as RT-PCR or RNAscope and by quantifying the expression of some YAP target’s genes (CTGF, Cyr61, Amphiregulin, Integrin B1, etc.). However, as the patients of this retrospective cohort have been operated for colon cancer at Caen University Hospital between January 2010 and September 2013 and since it is known that RNAs are very unstable in FFPE samples, it is unfortunately not possible to carry out these techniques on such old samples.

Further investigations are in progress concerning this correlation.

Results described in the last paragraph should be included.

As requested by reviewer, Tableau 3 and Figure 1 have been added with the results.

Reviewer 2 Report

The article entitled “Deciphering promoter hypermethylation of genes encoding for RASSF/Hippo pathway reveals the poor prognostic factor of RASSF2 gene silencing in colon cancers” in which authors Riffet et at., aimed to investigate the frequency of promoter hypermethylation of genes encoding the RASSF/Hippo pathway in colon cancers. Finally authors concluded that the RASSF2 hypermethylation is a frequent event and an independent poor prognostic factor in colon cancer.

This can be considered for publication after addressing my comment below:

  1. Results should be compared against the normal subjects.

Author Response

The article entitled “Deciphering promoter hypermethylation of genes encoding for RASSF/Hippo pathway reveals the poor prognostic factor of RASSF2 gene silencing in colon cancers” in which authors Riffet et at., aimed to investigate the frequency of promoter hypermethylation of genes encoding the RASSF/Hippo pathway in colon cancers. Finally authors concluded that the RASSF2 hypermethylation is a frequent event and an independent poor prognostic factor in colon cancer.

This can be considered for publication after addressing my comment below:

  1. Results should be compared against the normal subjects.

We would like to thank the reviewer#2 for his/her interests and valuable comments on the manuscript.

We understand the concern of the reviewer#2. However, the previous studies by Hesson and Park have already reported the absence of methylation in the adjacent normal tissues of colonic cancer tissues. Indeed:

  • According to Hesson (https://doi.org/10.1038/sj.onc.1208566): “Methylation was observed in 21/30 (70%) tumours. Moreover, this methylation was always tumour-specific and was not detected in the matched patient’s DNA from normal mucosa (taken >10 cm from the primary).”
  • According to Park (https://doi.org/10.1002/ijc.22276.):” We also examined the methylation status in 17 corresponding normal tissues (Fig. 2a). MSP analysis showed that RASSF2A was unmethylated in most corresponding normal tissues, but 2 normal tissues showed weak methylation. These results were further confirmed by cloning and bisulfite sequencing; specifically, only 1–5 CpG sites of a total of 25 CpG sites were methylated in normal tissues (Fig. 2b).”

At the beginning of our study, we performed analyses on 5 normal mucosae far from the tumor. We found that these mucosae were all found to have an unmethylated promotor of the RASSF2 gene (data not shown). Because of the absence of RASSF2 methylation in normal colonic tissue, which has already been reported in the literature and of the absence of such hypermethylation in the 5 mucosa that we have tested, we did not further analyze RASSF2 methylation outside of the tumor (in healthy tissue) of all the patient of our cohort.

We now justify this in the Discussion section as follow:” Because of the absence of RASSF2 promoter hypermethylation in normal colonic tissue already reported in the literature [26, 28] and of the absence of such hypermethylation in five mucosa that we have tested at the beginning of this study (data not shown), we did not analyze RASSF2 promoter hypermethylation outside the tumor for all patient of our cohort.”

Reviewer 3 Report

This article describes analysis of clinical samples of colorectal cancers for the methylation status of components of the Hippo pathway. Using methylation specific PCR they analyse 229 patients representing different grades of cancer. They conclude that RASSF2 is the most interesting gene in their analysis; hypermethylation is associated with a poorer overall survival. Therefore, they propose clinical utility for therapeutic intervention.

This manuscript is well written and clearly outlines background, hypothesis and place in the wider literature. Results do support current knowledge. Patient consent has been gained. This paper is suitable for inclusion in the identified special issue.

Concerns

In the simple summary the authors claim that this is “the first report of a systematic analysis of the hippo pathway in colon cancer”. Yet in the discussion they compare their findings with other publications that have found similar methylation patterns for RASSF2, specifically Park et al (reference 23) and Hesson et al (reference 28). Therefore, this claim should be removed or further justification for this statement should be given. How is this data superior to the other studies mentioned? Due to this I could not give a higher novelty score.

In the methods section there is a large description of how the patient samples were analysed. MSI status is described – definition of MSI appears to be missing. The antibodies that were used are described, however there is no visual data in the paper relating to this. Furthermore, KRAS and BRAF mutation analysis is also described and again there is no visual data relating to this. This data should be included in the supplementary data so the reader can view it or as a minimum some representative examples. Data in a table should be supported with examples.

Figures

Figure 1A: An example of MS-PCR is shown. Why are there bands in the water control tracks? There bands are also present in the experimental lanes suggesting contamination. There is a clear lack of bands of the correct sizes in this control but it is not technically correct. Again, all PCR data should be provided in supplementary data so data robustness can be determined. 

Figure 1 C: The take home message of this paper is that HYPERMETHYLATED RASSF2 yields a poorer prognosis yet in this figure the opposite is shown. Hypermethylated RASSF2 provides greater patient survival. Is the graph incorrectly labelled?

This is why I have given a low score for scientific soundness. I would expect this to be improved significantly as revision.

Further suggested improvements

This paper would benefit from the inclusion of more data (which they have generated but not shown here). Perhaps a heat map type diagram depicting the data from each patient sample so that patients with more than one hypermethylated hippo component and/or mutation status can be observed by the reader. I feel that the authors have undersold themselves by not showing more of the actual data they have generated.

Author Response

This article describes analysis of clinical samples of colorectal cancers for the methylation status of components of the Hippo pathway. Using methylation specific PCR they analyse 229 patients representing different grades of cancer. They conclude that RASSF2 is the most interesting gene in their analysis; hypermethylation is associated with a poorer overall survival. Therefore, they propose clinical utility for therapeutic intervention.

This manuscript is well written and clearly outlines background, hypothesis and place in the wider literature. Results do support current knowledge. Patient consent has been gained. This paper is suitable for inclusion in the identified special issue.

We would like to thank the reviewer#3 for his/her interests and valuable comments on the manuscript.

Concerns:

In the simple summary the authors claim that this is “the first report of a systematic analysis of the hippo pathway in colon cancer”. Yet in the discussion they compare their findings with other publications that have found similar methylation patterns for RASSF2, specifically Park et al (reference 26) and Hesson et al (reference 28). Therefore, this claim should be removed or further justification for this statement should be given. How is this data superior to the other studies mentioned? Due to this I could not give a higher novelty score.

We thank the reviewer for pointing this out. The studies of Park and Hesson investigate only RASSF2 and not the whole RASSF/hippo pathway in tumors. Therefore, our study is “the first report of a whole systematic analysis of the hippo pathway in colon cancer”. The term “whole” has been added.

In the methods section there is a large description of how the patient samples were analysed. MSI status is described – definition of MSI appears to be missing. The antibodies that were used are described, however there is no visual data in the paper relating to this. Furthermore, KRAS and BRAF mutation analysis is also described and again there is no visual data relating to this. This data should be included in the supplementary data so the reader can view it or as a minimum some representative examples. Data in a table should be supported with examples.

We thank the reviewer for raising this critical issue. We have replaced the term of “MSI” by “Mismatch Repair status”, in the material and method section which is more appropriate and give more details on the protocol of the different immunostainings allowing to diagnose this status. You can find the modified text in the revised manuscript as follows: “MisMach Repair (MMR) status was analyzed by immunohistochemical technique with the following four antibodies: M1 (anti-MLH1), G219-1129 (anti-MSH2), 44 (anti-MSH6), and EPR3947 (anti-PMS2) on a Ventana Benchmark ULTRA automated system. In case of conserved nuclear expression throughout the tumor proliferation, the lesion was classified as proficient MMR in immunohistochemistry (pMMR-IHC). In case of extinction of one or two markers of the MMR system in immunohistochemistry, an additional microsatellite instability test was performed in molecular biology to confirm the deficient MMR (dMMR) status.”

In addition, we now added new illustrations of the different grades of differentiation of adenocarcinomas (Figure S1), the histological criteria of the cohort (Figure S2) and a deficient MMR (dMMR) status (Figure S3). Regarding to the KRAS and BRAF mutations, we cannot illustrate them since these analyzes were subcontracted by the “François Baclesse cancer center”, which provides us a result without graphic representation. However, these analyzes are very conventional in hospital management practices for patients with a colon cancer.

Figures:

Figure 1A (now 2A): An example of MS-PCR is shown. Why are there bands in the water control tracks? There bands are also present in the experimental lanes suggesting contamination. There is a clear lack of bands of the correct sizes in this control but it is not technically correct.

We thank the reviewer for pointing this out. The visible bands in the water control correspond to the primer dimers. The primers used in this study are between 22 and 28 bases (please refer to table S1 for more details), their dimers are around 50 bp, they correspond to the signals that we see at the bottom of each track of the agarose gel shown in this figure, it is thus not a contamination but the reflect of the excess of primers.

We understand that the illustration presented initially in the figure raised questions for reviewer, we therefore propose a new illustration of the MS-PCR of RASSF2 with more clear annotation to avoid any confusions for the readers.

Again, all PCR data should be provided in supplementary data so data robustness can be determined.

With all due respect for this reviewer, it is uncommon to publish the all PCR data, we hope that with the new figure of a typical RASSF2 promoter methylation test result, the reviewer will be more convinced of our data. We also take the liberty of indicating to this reviewer that the validation of these PCRs was previously published in Levallet et al, 2019 (DOI: 10.1016/j.jmoldx.2019.03.007).

In addition, we questioned the TCGA database to test if our results were found in another cohort. Analysis of the influence of the expression level of RASSF2 mRNA was performed in 438 patients with colon cancer. Patients were classified according to whether their tumor strongly (n= 336) or weakly (n=102) expresses the RASSF2 mRNA. Survival analysis reveals that the low RASSF2 mRNA expression (possibly due to hypermethylation of the RASSF2 promoter) predicts significant worse overall survival in 438 patients with colon cancer (P score: 0.038, Expression of RASSF2 in colorectal cancer - The Human Protein Atlas). We have added this information in the results of the revised manuscript, and in figure S5.

Figure 2 C: The take home message of this paper is that HYPERMETHYLATED RASSF2 yields a poorer prognosis yet in this figure the opposite is shown. Hypermethylated RASSF2 provides greater patient survival. Is the graph incorrectly labelled?

We thank the reviewer for pointing this out and we apologize for this carelessness. Indeed, there was an error in this the graph and the two legends were reversed.

This is why I have given a low score for scientific soundness. I would expect this to be improved significantly as revision.  

We understand this point of view and hope to have met the expectations of this reviewer.

Further suggested improvements

This paper would benefit from the inclusion of more data (which they have generated but not shown here). Perhaps a heat map type diagram depicting the data from each patient sample so that patients with more than one hypermethylated hippo component and/or mutation status can be observed by the reader. I feel that the authors have undersold themselves by not showing more of the actual data they have generated.

We thank this reviewer for this very interesting suggestion. Following its recommendation, two heat maps were added regarding the results of RASSF/hippo pathway actor methylation data as well as clinical and histological data (Figure 1 and Figure S4).

Round 2

Reviewer 1 Report

The authors have addressed all my concerns.

Reviewer 2 Report

Authors’ addressed the concern of this reviewer and it’s now suitable for publication in this journal. 

Reviewer 3 Report

Thank you for responding to my comments appropriately. I now feel that this study has appropriate robustness to be published.

There is a typographical error in Figure 1 legend "stagging" instead of staging.

I have looked to the publication which describes the RASSF2 primers. Indeed, they also have a problem with primer dimer but observe a different banding pattern. However, with the added annotations and the inclusion of validation in another data set I am satisfied that conclusions based on the data are genuine.